# FIRE-Bench: Evaluating Research Agents on the Rediscovery of Scientific Insights

## Abstract

Autonomous agents powered by large language models (LLMs) promise to accelerate scientific discovery, but rigorously evaluating their capacity for genuine discovery remains a critical challenge. Current evaluation benchmarks face a dilemma: they either rely on LLM-as-judge evaluations of auto-generated papers, which raise concerns about validity and circularity, or focus on optimizing single performance metrics that serve as a coarse proxy for genuine discovery. To address this, we introduce FIRE-Bench (**F**ull-cycle **I**nsight **R**ediscovery **E**valuation). Our benchmark reframes evaluation by tasking agents with the verifiable rediscovery of established scientific findings from recent, high-impact ML research. We provide agents only with the high-level research question from a published study, requiring them to autonomously design experiments, implement code, execute their plan, and derive a conclusion from the evidence. We evaluate a suite of state-of-the-art agents with frontier model backbones (e.g., GPT-5) on FIRE-Bench. Our findings paint a sobering picture of current capabilities: even the most advanced agents struggle profoundly, exhibiting low success rates, high variance, and a spectrum of recurring failure modes ranging from flawed experimental design to ungrounded conclusions. FIRE-Bench provides a rigorous, diagnostic framework for measuring and driving progress towards AI agents capable of genuine scientific discovery.

## 1 Introduction

The emergence of autonomous agents powered by large language models (LLMs) holds the promise of accelerating scientific discovery at an unprecedented scale. These "AI researchers" are increasingly capable of automating discrete stages of the research lifecycle, from literature synthesis Zheng et al. (2025); Schmidgall & Moor (2025), hypothesis generation Baek et al. (2024); Si et al. (2024), to coding Tian et al. (2024); Chan et al. (2024), experimentation Kon et al. (2025), and data analysis Majumder et al.; Gu et al. (2024). However, a fundamental challenge lies in rigorously evaluating their capacity for genuine scientific discovery. Validating novel outcomes often requires resource-intensive, real-world verification, such as wet-lab experiments or large-scale human expert studies. This evaluation bottleneck becomes particularly acute for agents designed to automate the full research cycle—from an initial question to a final, empirically-grounded conclusion Lu et al. (2024); Yamada et al. (2025); Schmidgall et al. (2025). Assessing the validity and integrity of an entire research trajectory, *rather than an isolated component*, presents a far more complex and multifaceted challenge.

Current benchmarks for full-cycle research agents largely follow two distinct paradigms. The first, and more ambitious, paradigm evaluates agents for generating entire research papers from a high-level research topic Lu et al. (2024); Yamada et al. (2025); Schmidgall et al. (2025). However, rigorously assessing the scientific merit of these artifacts is a fundamental bottleneck. While human peer review offers a gold standard sometimes, it is prohibitively slow and expensive for large-scale benchmarking. Therefore, many efforts resort to using the LLM-as-judge for evaluation—an approach fraught with concerns about reliability, bias, and circular reasoning, making it an inadequate proxy for rigorous scientific validation Zheng et al. (2023); Schroeder & Wood-Doughty (2024); Ye et al. (2024). The second paradigm avoids subjective paper evaluation by focusing on ML tasks with a single, objective performance metric, such as improving model accuracy on a leaderboard Huang et al. (2024b); Chan et al. (2024); Wijk et al. (2024). While new methods can emerge (Zhang et al., 2025)), these

benchmarks often measure rigid replication (Starace et al., 2025) and engineering proficiency (Chan et al., 2024). Although objective, their reliance on numerical performance metrics provides a coarse-grained signal that overlooks the crucial, nuanced process of scientific reasoning and thus provides fewer insights for agent behaviors.

To bridge this critical evaluation gap, we introduce FIRE-BENCH (**F**ull-cycle **I**nsight **R**ediscovery **E**valuation), a benchmark designed to rigorously evaluate a research agent's ability to conduct a full cycle of empirical research and arrive at a verifiable scientific conclusion. Our core principle is to reframe the evaluation of discovery: instead of tasking agents with generating novel, unverified claims (or papers), we assess their ability to autonomously rediscover established, non-trivial insights from recent machine learning literature. FIRE-BENCH is constructed from impactful analysis papers focusing on LLM behaviors, where the central findings are empirical, well-documented, and computationally verifiable. Crucially, while we use existing findings as ground truth, FIRE-BENCH is *distinct from a simple reproducibility task*–given only a high-level research question from a source paper, we systematically mask the original authors' methodology, experimental design, and analytical path. This formulation creates a constrained yet open-ended discovery problem where the agent must independently hypothesize, plan, experiment, and analyze results to reach the target insight. This approach offers two key advantages: it grounds evaluation in *an objective, human-validated scientific truth*, obviating the need for unreliable LLM judges, while simultaneously enabling *a fine-grained analysis* of the agent's scientific reasoning process, moving far beyond a single, coarse-grained performance metric.

FIRE-BENCH is designed to provide the fine-grained analysis that current methods lack, dissecting agent capabilities across four core stages of research: *Research Planning, Implementation, Experimental Execution, and Empirical Analysis*. To quantify the final success, we measure the claim-level $F_1$ score between an agent's synthesized conclusions and the ground-truth find-

Figure 1: Comparison of benchmark paradigms.

| | **Full Cycle** | **Insight Driven** | **Grounded Eval.** | **Method. Explor.** |
|---|---|---|---|---|
| Method Repl./Eng. | ✓ | ✗ | ✓ | ✗ |
| Isolated Stage Auto. | ✗ | ✗ | ✓ | ✗ |
| Full Paper Gen. | ✓ | ✓ | ✗ | ✓ |
| **FIRE-Bench** | ✓ | ✓ | ✓ | ✓ |

ings. We evaluate a suite of state-of-the-art agents, including OpenHands Wang et al. (2025) and Claude Code, powered by frontier models such as GPT-5 and Claude-4-Sonnet. Our extensive experiments paint a sobering picture: even the most capable contemporary agents struggle significantly with full-cycle scientific inquiry. Performance is generally low and exhibits high variance across runs, indicating a lack of reliability. Our analysis reveals a diverse array of failure modes spanning the entire research pipeline, from flawed initial planning and incorrect code implementation to premature task termination and conclusions ungrounded in empirical evidence. These results highlight the profound challenges that remain and establish FIRE-BENCH as a crucial diagnostic tool for measuring and advancing the scientific reasoning capabilities of AI agents.

## 2 RELATED WORK

Numerous benchmarks for AI research agents have emerged. While agents are being developed for domains like chemistry and biology Swanson et al. (2024); Bran et al. (2023); M. Bran et al. (2024), our work is scoped to automating machine learning research. Existing benchmarks can be categorized by the breadth of the research cycle they evaluate.

**Benchmarks for Fragmented Research Stages**. A significant body of work assesses agent capabilities on discrete, isolated stages of the scientific research workflow. In the early stages, benchmarks like ResearcherBench (Xu et al., 2025) and DeepResearchBench (Du et al., 2025) evaluate an agent's ability to conduct deep literature synthesis. For idea generation, benchmarks such as IdeaBench (Guo et al., 2025) and ResearchBench (Liu et al., 2025) assess the novelty and feasibility of agent-proposed hypotheses. The execution phase, particularly coding and data analysis, has received the most attention. Benchmarks like SciCode (Tian et al., 2024) focus on general scientific coding, while others like BLADE (Gu et al., 2024), DiscoveryBench (Majumder et al.), and ScienceAgentBench (Chen et al.) specifically target agents' abilities in post-hoc data analysis and hypothesis testing. While valuable

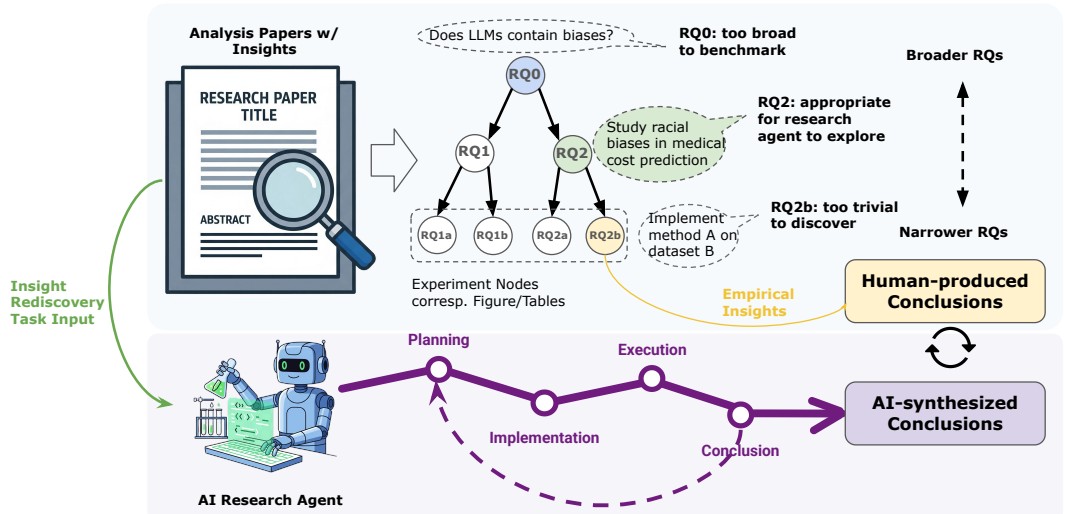

Figure 2: FIRE-BENCH presents an AI research agent with a high-level question from a seminal paper and evaluates its ability to autonomously rediscover the paper's core insight, enabling a fine-grained comparison of the entire machine-generated research process against the original human workflow.

for measuring specific competencies, these benchmarks do not assess an agent's ability to integrate these skills across the entire research cycle—a prerequisite for genuine end-to-end discovery.

**Benchmarks for Full-Cycle Research**. More ambitious benchmarks that target the full research cycle fall into two main paradigms. Metric-Driven Discovery. One paradigm tasks agents with improving a quantitative metric on a competitive task. Benchmarks like MLAgentBench (Huang et al., 2024b), MLE-Bench (Chan et al., 2024), and MLRC-Bench (Zhang et al., 2025) evaluate agents on engineering challenges or leaderboard-driven method discovery. However, their reliance on a single performance metric offers a coarse-grained signal that often prioritizes engineering skill over scientific reasoning. Automated Paper Generation. A second paradigm tasks agents with generating entire research papers from a prompt, as seen in The AI Scientist (Lu et al., 2024) and Agent Laboratory (Schmidgall et al., 2025). This approach faces a critical evaluation bottleneck. Human peer review is prohibitively expensive for large-scale assessment, leading many to use LLM-as-judge systems (Lu et al., 2024; Yamada et al., 2025; Weng et al., 2024). This alternative is widely criticized for its unreliability and potential for circular reasoning, making it unsuitable for rigorous scientific validation. FIRE-BENCH avoids both pitfalls by grounding evaluation in verifiable, established insights.

**Benchmarks for Scientific Reproducibility**. Our "rediscovery" paradigm shares a conceptual similarity with reproducibility benchmarks like PaperBench (Starace et al., 2025) and LMR-Bench (Yan et al., 2025), which also leverage existing publications as ground truth. These benchmarks assess an agent's ability to replicate the experiments and results described in a given paper. However, a critical distinction lies in the problem formulation. Reproducibility tasks typically provide the agent with the full context of the paper, including the detailed methodology and expected outcomes. In contrast, FIRE-BENCH provides only the high-level research question, intentionally masking the original experimental design, implementation details, and analytical path. This transforms the task from one of replication to one of constrained, open-ended discovery, compelling the agent to demonstrate genuine scientific reasoning rather than translating a prescribed procedure into code.

## 3 FIRE-BENCH: FROM PAPERS TO VERIFIABLE DISCOVERY TASKS

### 3.1 BENCHMARK CONSTRUCTION

Our methodology for constructing FIRE-BENCH is centered on a principled process of research-problem decomposition. This process transforms high-quality, empirical analysis papers into verifi-

able benchmark tasks that balance exploratory freedom (i.e., *too broad to benchmark*) with empirical verifiability (i.e., *too narrow to explore*).

**Research-Problem Tree Abstraction** We formalize the intellectual structure of a given empirical analysis paper $\mathcal{P}$ as a hierarchical **research-problem tree**, denoted by $\mathcal{T}(\mathcal{P})$. This abstraction explicitly encodes the reasoning trajectory of the original authors, from a broad, overarching research question (e.g., *"Do this LLM contain biases?"*) to specific experimental tasks used to empirically validate insights (e.g., *"To verify racial biases, implement a method to ..."*).

Formally, the research-problem tree $\mathcal{T}(\mathcal{P})$ consists of three types of nodes:

- **Root Node** ($r$): The root node encapsulates the broadest research problem addressed by $\mathcal{P}$, derived directly from prominent textual cues, typically in the title, abstract, introduction, or discussion.
- **Intermediate Nodes** ($v_i \in \mathcal{V}_I$): Each intermediate node represents a narrower subproblem explicitly introduced by the original authors as a logical step toward addressing the root. Intermediate nodes can recursively decompose into finer-grained subproblems until reaching leaf nodes.
- **Leaf Nodes** ($l_j \in \mathcal{L}$): Each leaf node constitutes a fully specified experimental task, precisely detailing dataset ($\mathcal{D}_j$), model or method ($\mathcal{M}_j$), evaluation metrics ($\mathcal{C}_j$), and experimental protocols ($\mathcal{E}_j$). Crucially, each leaf maps directly to explicit results in $\mathcal{P}$ (figures, tables, or result sections), thereby ensuring verifiable grounding.

**Automated Research-Problem Tree Extraction** To operationalize the extraction of $\mathcal{T}(\mathcal{P})$, we introduce an automated parsing procedure. Specifically, we instantiate a research-problem parser $E_\phi$, a deterministic prompting approach parameterized by a frontier LLM, GPT-5 Pro (one of the strongest LLMs available during our experimentation):

$$E_\phi : \Sigma^* \to \mathcal{T}, \quad \mathcal{T}(\mathcal{P}) = E_\phi(\mathcal{P}) \tag{1}$$

Prior research has demonstrated that advanced LLMs, with carefully engineered prompts, are capable of reliably parsing complex documents and extracting structured hierarchical representations (Ma et al., 2024). The robustness of $E_\phi$ was further verified by human expert inspection, ensuring accurate reflection of the authors' original reasoning processes. Full prompt details available in Appendix B.

**Selecting Research Problems for Evaluation** From the generated research-problem tree, we formulate a benchmark task by sticking with the paper's key empirical insights. First, we identify a target leaf node ($l^* \in \mathcal{L}$) that corresponds to a central finding, usually corresponding to the main figure or table. We then trace the unique logical path from the root node ($r$) to this leaf.

We then choose an appropriate intermediate node $v^* \in \mathcal{V}_I$. This node is strategically chosen to balance exploratory openness, allowing genuine discovery opportunities for research agents, and concreteness, providing a clearly defined scope suitable for empirical validation.

By selecting a node mid-path carefully, we create a well-defined **"constrained rediscovery" problem**. The agent receives only the question from $v^*$, while the original author's methodology and the conclusion from the target leaf $l^*$ are withheld. This conclusion from $l^*$ serves as the ground truth for our evaluation. We also format the research input nicely, an example presented in Appendix B.

## 3.2 EVALUATION PROTOCOL

We evaluate an agent's performance by comparing its final synthesized conclusion against the ground-truth findings from the source paper. Adopting the rigorous framework of *RAGChecker* (Ru et al., 2024), we perform a fine-grained, claim-level analysis.

Both the agent's conclusion and the ground-truth text are decomposed into sets of atomic, verifiable claims, denoted $C_{agent}$ and $C_{gt}$, respectively. This process is automated via LLM inference, with prompts provided in Appendix B. We then compute standard metrics to assess the quality of the agent's rediscovered insights:

$$\text{Precision} = \frac{|C_{\text{agent}} \cap C_{\text{gt}}|}{|C_{\text{agent}}|}, \quad \text{Recall} = \frac{|C_{\text{agent}} \cap C_{\text{gt}}|}{|C_{\text{gt}}|}, \quad F_1 = 2 \cdot \frac{\text{Precision} \cdot \text{Recall}}{\text{Precision} + \text{Recall}} \tag{2}$$

This meticulous evaluation protocol allows FIRE-BENCH to provide comprehensive insights into the nuanced scientific reasoning capabilities of contemporary research agents, which we will show in our systematic error analysis. Additional implementation details, including exact prompts, are provided in Appendix B.

### 3.3 SOURCE PAPER SELECTION AND FILTERING

The quality of **FIRE-Bench** depends critically on the source papers. We curated a set of high-impact, empirical analysis papers on LLM behavior from top-tier conferences (ICLR, ICML, NeurIPS) in 2024 and 2025. The full list of papers is available in Table 3.

Our selection process involved a multi-stage filtering protocol to identify suitable candidates. After an initial keyword-based search for "LLM" and "Language Model," we used an LLM-based classifier to identify papers whose primary contribution is the *analysis of LLM behaviors*, filtering out those focused on new model architectures, training techniques, or purely theoretical contributions. Finally, the authors manually reviewed the candidates to ensure they met three essential criteria for creating fair and feasible rediscovery tasks:

- **Open Inputs**: The research must rely on publicly available data and models, with no proprietary assets required for replication.
- **Compute-Light Execution**: The core experiments must be computationally tractable, runnable within a few hours on modest hardware, precluding tasks that require large-scale model training.
- **Non-trivial, Verifiable Insights**: The central findings must be specific, empirical claims clearly supported by figures or tables in the paper, not general platitudes.

## 4 EXPERIMENTS

**Agent Frameworks& LLMs**. We evaluate three state-of-the-art coding agents with different LLM backbones on FIRE-BENCH. Specifically, we include *OpenHands* (Wang et al., 2025), an open-source multi-agent system designed for autonomous software development. It is built on the CodeAct architecture (Wang et al., 2024) and augmented with additional agents for sub-tasks, like information gathering and step-level evaluation, as well as specialized tools. For further details, we refer readers to the corresponding paper and code repository.[1] For OpenHands, we experiment with both `gpt-o4-mini` and `gpt-5`. To ensure a comprehensive comparison, we also include two proprietary subscription-based agents: OpenAI's *Codex* and Anthropic's *Claude Code*. Each is evaluated with its default LLM, namely `gpt-5-medium` for *Codex* and `Claude-4-Sonnet` for *Claude Code*.[2] While the implementation details of proprietary agents (e.g., *Claude Code*) are not publicly available, we ensure that all agents have access to necessary tools, such as shell execution and file operation, for task execution.

**Experimental Details.** We run each agent in a sandbox environment via its Command-Line Interface (CLI). The sandbox is hosted on a GPU node with eight 80GB A100 GPUs, and all necessary API keys are configured locally. Each agent's working directory contains an instruction file specifying the task information (e.g., research question and experimental constraints, as described in §3), along with the provided datasets. We do not preconfigure additional environments (e.g., installing Python packages), as we regard such setup as part of the agents' capabilities. Trajectory analysis confirms that current coding agents are capable of handling these setup tasks effectively. A potential concern is that agents might attempt to retrieve the original paper via web search instead of generating their own experimental plan. However, trajectory inspection shows that agents consistently followed our instructions for experimental exploration.[3]

Each task-agent pair is executed three times to assess reproducibility, and we report the mean performance along with standard deviation. No hard runtime limit is imposed, though most runs complete within one hour. For evaluation, we adopt the *RAGChecker* library,[4] using its default models: `gpt-4o` for claim extraction and `gpt-4o-mini` for claim verification.

---

[1]https://github.com/All-Hands-AI/OpenHands

[2]Experiments were conducted primarily in August 2025; default checkpoints for proprietary agents may change over time. We adopt the defaults to reflect their optimized settings.

[3]One possible mitigation, as explored in Starace et al. (2025), is to blacklist specific webpages within the agent's browsing tool. In practice, we observed no such behavior, and agents adhered to our experimental instructions. Moreover, implementing blacklists is technically infeasible for proprietary agents. A systematic treatment of this issue is left to future work.

[4]https://github.com/amazon-science/RAGChecker

Table 1: Performance comparison across tasks. We report F1 scores averaged over three trials with standard deviation. Best results for each task are shown in **bold**.

| Task | OpenHands (o4-mini) | OpenHands (gpt-5) | Codex CLI (gpt-5-medium) | Claude Code (Sonnet-4) |
|---|---|---|---|---|
| Lost in the Middle | 57.0±49.6 | 71.1±7.7 | **91.7±14.4** | 60.1±30.0 |
| LLM Racial Bias in Medicine | **34.2±31.7** | 0.0±0.0 | 10.5±18.2 | 0.0±0.0 |
| LLMs Lack Self-Correction | 20.0±34.6 | 26.7±23.1 | 13.3±23.1 | **42.6±10.7** |
| Awareness Detection | 31.5±27.4 | 20.0±17.6 | 10.3±17.8 | **66.7±57.7** |
| CoT Faithfulness Gaps | 20.5±35.5 | 61.6±41.1 | **72.7±23.0** | 66.7±28.9 |
| CoT Without Prompting | 16.7±28.9 | 26.4±28.0 | 13.8±23.9 | **82.6±23.0** |
| Hallucination Snowballing | 58.0±50.8 | 69.2±19.5 | **80.9±21.8** | 77.6±4.3 |
| Counterfactual Simulatability | 41.4±20.0 | 44.0±15.7 | 0.0±0.0 | **39.2±31.7** |
| Premise Order Effects | 72.5±16.2 | **79.6±26.3** | 56.7±20.0 | 33.3±57.7 |
| Persona Reasoning Biases | 18.5±17.0 | 17.4±15.3 | **57.0±12.0** | 54.8±20.4 |
| MCQ Selection Bias | 40.7±27.1 | 51.3±22.8 | 59.2±20.0 | **62.9±7.4** |
| Prompt Formatting Sensitivity | 25.7±23.5 | 26.2±24.0 | **32.7±10.0** | 45.5±10.0 |
| Space-Time Representations | 42.3±20.3 | 46.2±13.1 | 33.6±12.8 | **51.5±17.0** |
| LLM Confidence Elicitation | 10.8±18.7 | 28.6±8.6 | 17.9±19.4 | **33.1±21.7** |
| ICL from Repetition | 32.5±30.0 | **57.3±6.0** | 55.4±7.9 | 34.3±30.8 |
| **Average (All Tasks)** | 34.8±28.8 | 41.7±17.9 | 40.4±16.3 | **50.1±23.4** |

# 5 RESULTS & ANALYSES

## 5.1 MAIN RESULTS

Our experiments reveal the profound difficulty of full-cycle research automation. As shown in Table 1, the performance of even the most advanced agents is low and inconsistent, underscoring the significant gap between current capabilities and the demands of genuine scientific inquiry.

**Overall Performance is Low and Highly Unreliable.** The most striking finding is the poor overall performance across the board. The best-performing agent, Claude Code, achieves an average $F_1$ score of only 49.3, while others lag behind. This sobering result indicates that reliably rediscovering non-trivial scientific insights remains beyond the grasp of current systems. Furthermore, the extremely high standard deviations for nearly all agent-task pairs highlight a critical lack of reliability. For instance, on *Lost in the Middle*, OpenHands (o4-mini) scores 57.0±49.6, and on *Awareness Detection*, Claude Code scores 66.7±57.7. This variance suggests that an agent's research trajectory is highly sensitive to stochastic factors in the LLM's generation process. Success often appears to be a "lottery," making these agents untrustworthy for tasks where correctness and reproducibility are paramount.

**Performance Varies with Conceptual Complexity.** Agent success is strongly correlated with the conceptual structure of the research task.

- **Success on Linear Tasks:** Agents perform best on tasks that admit a relatively linear and direct solution path. For example, in *Lost in the Middle* (best score: 91.7) and *Hallucination Snowballing* (best score: 80.9), the objective is clear and the experimental procedure is straightforward. In these cases, the task reduces to a complex but well-defined engineering problem, where agents excel.
- **Failure on Conceptually Nuanced Tasks:** Conversely, performance degrades sharply on problems that demand creative decomposition or a deep conceptual understanding. A prime example is *LLM Racial Bias in Medicine*. This task requires a nuanced causal experiment: first designing a bias-free control by removing racial indicators, then selectively re-introducing them to isolate their effect. **Critically, every agent failed to devise this control-based methodology.** They instead injected race information directly, a flawed approach that conflates correlation with causation and fails to

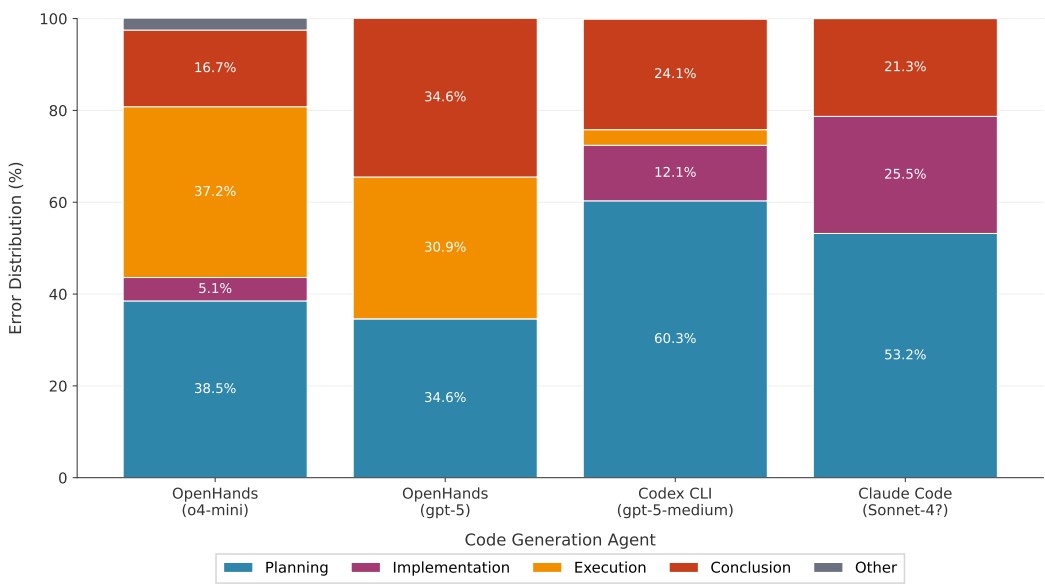

Figure 3: Error analysis with error distribution stacked per model.

rediscover the paper's core insight. This pattern demonstrates a fundamental weakness in abstract experimental design.

**Claude Code and Frontier Models Lead, But Gaps Remain.** Among the systems tested, Claude Code (`Sonnet-4`) is the strongest performer, achieving the highest average score and securing the best result on 7 of the 15 tasks. The proprietary agents, including Codex CLI (`gpt-5-medium`), generally outperform the open-source OpenHands framework. Within OpenHands, the choice of backbone model is crucial: upgrading from `gpt-o4-mini` to `gpt-5` yields a substantial average $F_1$ improvement of +10.3 points (from 31.4 to 41.7). This underscores that the advanced reasoning capabilities of frontier models are essential, though still insufficient, for tackling these complex research challenges.

## 5.2 FINE-GRAINED ERROR ANALYSIS

To gain deeper insight into agent performance, we conduct a claim-level error analysis. Specifically, we trace false negatives and false positives back to four stages of the exploration process, as identified from the agent logs: *Planning*, *Implementation*, *Execution*, and *Conclusion*. Errors at each stage are further categorized into representative types. For example, errors in *Planning* are divided into *Faulty Plan* (agents omit or mis-specify essential steps) and *Off-Target* (agents deviate from the intended research goal). In total, we define nine categories across the four stages. Definitions and examples of each error type are provided in the Appendix 5. Manual inspection of agent logs is challenging due to their length and complexity. To address this, we adopt a hybrid human-LLM approach. For each erroneous claim, the corresponding log is provided to an LLM, which predicts the error stage, assigns an error type, and generates a rationale that specifies the location of the error in the log. These outputs are then reviewed by the authors to verify the reliability of the error tracing.

This fine-grained analysis, summarized in Figure 3, reveals starkly different failure profiles for each agent, pointing to fundamental differences in their architectural strengths and weaknesses.

**Proprietary Agents Fail at Planning, Not Execution.** The most striking pattern is that the top-performing agents, Codex CLI and Claude Code, exhibit a failure profile dominated by the Planning stage. A staggering 60.3% of errors for Codex CLI and 53.2% for Claude Code originate from flawed or incomplete initial plans. These agents often misunderstood the core objective, failed to design necessary control conditions, or omitted critical steps entirely. However, once a plan was formulated (even a flawed one), they excelled. Both agents had minimal Execution errors (3.4% and 0.0%, respectively), indicating robust capabilities in running code, debugging, and handling the

Table 2: Average cost across tasks

| Task | OpenHands (o4-mini) | OpenHands (gpt-5) | Codex CLI (gpt-5-medium) | Claude Code (Sonnet-4) |
|---|---|---|---|---|
| Lost in the Middle | 0.4283 | 0.9028 | 0.1667 | 0.8120 |
| LLM Racial Bias in Medicine | 0.6915 | 0.5117 | 0.1467 | 0.9268 |
| LLMs Lack Self-Correction | 0.4311 | 1.0094 | 0.1600 | 1.5578 |
| Awareness Detection | 0.7976 | 1.3780 | 0.1933 | 0.8428 |
| CoT Faithfulness Gaps | 0.2761 | 0.6520 | 0.2700 | 0.7118 |
| CoT Without Prompting | 1.2083 | 0.3934 | 0.3700 | 1.0843 |
| Hallucination Snowballing | 0.5873 | 0.8342 | 0.1285 | 0.7931 |
| Counterfactual Simulatability | 0.3729 | 0.4587 | 0.0714 | 0.4026 |
| Premise Order Effects | 0.5128 | 0.6475 | 0.1027 | 0.5389 |
| Persona Reasoning Biases | 0.4217 | 0.6124 | 0.1098 | 0.9945 |
| MCQ Selection Bias | 0.3835 | 0.4372 | 0.0843 | 0.6510 |
| Prompt Formatting Sensitivity | 0.5111 | 0.6589 | 0.0646 | 0.7136 |
| Space-Time Representations | 0.7061 | 0.8473 | 0.0484 | 0.5447 |
| LLM Confidence Elicitation | 0.2310 | 0.3186 | 0.2330 | 1.3153 |
| ICL from Repetition | 1.3403 | 1.0824 | 0.0596 | 0.7779 |
| **Total** | **8.8996** | **10.7445** | **2.2090** | **12.6671** |

environment. This suggests that the primary bottleneck for frontier agents is not the mechanical act of coding, but the higher-level cognitive task of scientific experimental design.

**OpenHands Shows Divergent and More Varied Failure Modes.** The OpenHands agents present a more varied and less mature error profile.

- **OpenHands (`o4-mini`)**, powered by a smaller model, struggles across the entire research pipeline. Its errors are distributed almost evenly across Planning (38.5%) and Execution (37.2%), with a significant portion in Conclusion (16.7%). This profile is characteristic of a less capable agent that is brittle at every stage, from initial strategy to final interpretation.

- **OpenHands (`gpt-5`)**, despite its stronger backbone model, surprisingly makes the most errors overall (81). While it improves significantly on planning compared to its predecessor, its failures become concentrated in the later stages of Execution (30.9%) and, most notably, Conclusion (34.6%). This high rate of conclusion-related errors is unique among the agents and points to a critical weakness in data interpretation. The agent successfully executes the experiment but then fails to correctly analyze the results to draw the right scientific insight.

## 5.3 COST-TIME ANALYSIS

Beyond raw performance, the practical viability of research agents hinges on their operational cost. We analyze the monetary expenditure for each agent across all tasks, with costs derived from API billing. For Codex CLI, which only reports token usage, we estimate costs based on the public pricing for its underlying model (`gpt-5-medium` as of September 2025), assuming a 3:1 input-to-output token ratio observed during our experiments. The results, summarized in **Table 2**, reveal a clear but complex relationship between cost and scientific capability.

**Higher Cost for Higher Performance.** Our primary finding is a strong correlation between an agent's cost and its performance. The top-performing agent, Claude Code (49.3 avg. $F_1$), is also the most expensive, with a total cost of $12.67. This trend is further confirmed within the OpenHands framework, where a controlled comparison is possible. Upgrading the backbone model from `gpt-o4-mini` to the more powerful `gpt-5` increases the cost by 20% (from $8.90 to $10.74) but yields a substantial 33% relative improvement in performance (from 31.4 to 41.7 avg. $F_1$). This

demonstrates that access to frontier reasoning capabilities, while effective, comes at a premium and is a key driver of both success and expenditure.

**Maximum Performance per Dollar.** While the general trend holds, Codex CLI stands out as a remarkable efficiency outlier. It is by far the most economical agent, with a total cost of only \$2.21—over 5 times cheaper than Claude Code. Despite this frugality, it achieved a competitive average $F_1$ score of 40.7, nearly matching the far more expensive OpenHands (`gpt-5`) agent. This makes Codex CLI the most cost-effective solution in our evaluation, suggesting that its internal architecture may be highly optimized for generating concise, effective action plans that minimize token consumption and expensive correction loops.

**Task Complexity as a Cost Driver.** On a per-task basis, costs often reflect the problem's intrinsic difficulty. Tasks requiring intricate, multi-step reasoning, such as *LLMs Lack Self-Correction* (\$1.56 on Claude Code) and *ICL from Repetition* (\$1.34 on OpenHands), consistently incurred higher costs across agents. This suggests that these problems demand longer reasoning chains, more tool interactions, or more trial-and-error, all of which translate directly to increased API usage. In conclusion, while a cost-performance trade-off currently defines the landscape, the notable efficiency of certain agents suggests that future architectural innovations could unlock top-tier performance without incurring prohibitive costs.

## 6 CONCLUSIONS

In this work, we addressed the critical challenge of rigorously evaluating autonomous agents for full-cycle scientific research. We identified fundamental limitations in existing paradigms, which either rely on unreliable metrics like LLM-as-judge or focus on coarse-grained performance outcomes that offer little scientific insight. To bridge this gap, we introduced FIRE-Bench, a benchmark built on the principle of **insight rediscovery**. By tasking agents to autonomously reproduce established findings from recent, high-impact ML papers, given only a high-level research question, our methodology grounds evaluation in objective, human-validated truth while enabling a fine-grained analysis of the entire research process.

Our extensive evaluation of state-of-the-art agents paints a clear and sobering picture: full-cycle scientific inquiry remains largely an unsolved problem. Overall performance is low, and high variance across runs highlights a critical lack of reliability. Crucially, our detailed error analysis reveals that as underlying models become more powerful, the primary bottleneck in scientific automation is shifting. For the most capable agents, failure is no longer dominated by low-level code implementation or execution errors, but rather by deficiencies in high-level cognitive tasks: flawed initial **planning** and the inability to draw correct, empirically-grounded **conclusions** from experimental results.

FIRE-Bench provides the community with a much-needed diagnostic tool to move beyond simple performance metrics and begin addressing these deeper challenges in scientific reasoning. Our findings suggest that future progress will depend less on incremental improvements in coding capabilities and more on fundamental advances in strategic planning, experimental design, and causal inference. The ultimate goal is not merely to build agents that can execute experiments, but agents that can reason like scientists. We hope our work will help steer research in this essential direction.

## LIMITATION

While FIRE-Bench offers a rigorous evaluation framework, we acknowledge its principal limitations. Our benchmark is designed to evaluate an agent's ability to solve a given research problem with a known solution. This "insight rediscovery" paradigm provides objective ground truth but does not assess the more abstract challenge of formulating novel research questions, a process requiring scientific "taste", nor does it capture true open-ended discovery where an agent might uncover a valid but unexpected finding. Evaluating these more creative facets of science remains a fundamental open problem.

Furthermore, the benchmark is currently focused on a specific sub-domain: computationally lightweight, empirical analysis of LLMs. This deliberate choice ensures tractability but means our findings on agent capabilities and failure modes may not generalize to other scientific domains, such as computational biology, or to research requiring the management of long-running, resource-intensive experiments. We view these boundaries not as shortcomings, but as a clear roadmap for the next generation of benchmarks and agents in scientific AI.

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

# A  PAPER SUMMARY

Table 3: Summary of Papers in FIRE-Bench

| # | Paper Title | Short Name | Venue | Reference |
|---|---|---|---|---|
| 1 | Unmasking and quantifying racial bias of large language models in medical report generation | *LLM Racial Bias in Medicine* | Nature Comm. Med. | Yang et al. (2024) |
| 2 | Lost in the Middle: How Language Models Use Long Contexts | *Lost in the Middle* | TACL, Vol. 12 | Liu et al. (2024) |
| 3 | Large Language Models Cannot Self-Correct Reasoning Yet | *LLMs Lack Self-Correction* | ICLR 2024 | Huang et al. (2024a) |
| 4 | Large Language Models Often Know When They Are Being Evaluated | *Awareness Detection* | arXiv | Needham et al. (2025) |
| 5 | Reasoning Models Don't Always Say What They Think | *CoT Faithfulness Gaps* | arXiv / Anthropic | Chen et al. (2025) |
| 6 | Chain-of-Thought Reasoning Without Prompting | *CoT Without Prompting* | NeurIPS 2024 | Wang & Zhou (2024) |
| 7 | How Language Model Hallucinations Can Snowball | *Hallucination Snowballing* | ICML 2024 | Zhang et al. (2024) |
| 8 | Do Models Explain Themselves? Counterfactual Simulatability of Natural Language Explanations | *Counterfactual Simulatability* | ICML 2024 | Chen et al. (2024b) |
| 9 | Premise Order Matters in Reasoning with Large Language Models | *Premise Order Effects* | ICML 2024 | Chen et al. (2024a) |
| 10 | Bias Runs Deep: Implicit Reasoning Biases in Persona-Assigned LLMs | *Persona Reasoning Biases* | ICLR 2024 | Gupta et al. (2024) |
| 11 | Large Language Models Are Not Robust Multiple Choice Selectors | *MCQ Selection Bias* | ICLR 2024 | Zheng et al. (2024) |
| 12 | Quantifying Language Models' Sensitivity to Spurious Features in Prompt Design | *Prompt Formatting Sensitivity* | ICLR 2024 | Sclar et al. (2024) |
| 13 | Language Models Represent Space and Time | *Space–Time Representations* | ICLR 2024 | Gurnee & Tegmark (2024) |
| 14 | Can LLMs Express Their Uncertainty? An Empirical Evaluation of Confidence Elicitation in LLMs | *LLM Confidence Elicitation* | ICLR 2024 | Xiong et al. (2024) |
| 15 | Understanding In-Context Learning from Repetitions | *ICL from Repetition* | ICLR 2024 | Yan et al. (2024) |

Table 4: Research Questions in FIRE-Bench Papers

| # | Short Name | The Core Question of Research Input |
|---|---|---|
| 1 | *LLM Racial Bias in Medicine* | Does the GPT-3.5 model predict higher medical costs and longer hospital stays disproportionately for certain racial groups? |
| 2 | *Lost in the Middle* | How does model performance vary based on relevant information position in context? |
| 3 | *LLMs Lack Self-Correction* | How do self-correction methods impact large language model performance across math, commonsense reasoning, and multi-hop question answering benchmarks? |
| 4 | *Awareness Detection* | To what extent can frontier language models detect that a given interaction transcript comes from an evaluation rather than real-world deployment, when tested across diverse chat settings? |
| 5 | *CoT Faithfulness Gaps* | To what extent do reasoning models' chains-of-thought faithfully reflect their internal reasoning processes when they exploit external hints? |
| 6 | *CoT Without Prompting* | Can large language models, without any chain of thought prompts, reveal reasoning paths and improve answer accuracy by altering its decoding approach? |
| 8 | *Counterfactual Simulatability* | Do natural language explanations provided by language models enable humans to accurately simulate the model's behavior under counterfactual inputs? |
| 9 | *Premise Order Effects* | Does the order of premises affect the reasoning performance of LLMs, even when the logical content remains the same? |
| 11 | *MCQ Selection Bias* | Are modern large language models (LLMs) robust in handling multiple choice questions (MCQs), and if not, what causes their vulnerability, especially regarding their sensitivity to option position changes, and how can such issues be mitigated? |
| 13 | *Space–Time Representations* | Do large language models (LLMs) learn more coherent and grounded representations that reflect the real world (such as spatial and temporal representations) rather than just an enormous collection of superficial statistics? |
| 15 | *ICL from Repetition* | What is the underlying mechanism of in-context learning (ICL) in Large Language Models (LLMs), and how do surface repetitions, particularly token co-occurrence reinforcement, influence ICL, including both its beneficial functions and detrimental effects? |

## B    PROMPTS USED IN FIRE-BENCH

---

**Paper Parsing Prompt**

```
You are a research-paper expert specializing in methodological
    analysis and problem decomposition of scientific studies.

**GOAL**
- Fully comprehend the paper, understand its core research problems
    and experiments
- Then, parse the given paper and construct a hierarchical **
    research-problem tree** that mirrors the authors logic as
    follows:

* **Root node**  the single, more essential, broadest research
    problem tackled by the paper.
* **Intermediate nodes**  progressively narrower sub-problems/
    questions/objectives that the authors introduce to tackle the
    root.
* **Leaf nodes**  fully specified experimental tasks (datasets,
    models, metrics, or protocols) that map to a *figure, table, or
    named result section* in the paper.

Continue decomposing until every branch ends in such a leaf. There
    is no depth limit.

---

###  Reading & Extraction Rules
1. **Locate the root** in the title, abstract, introduction, or
    discussion.
2. **Recursively decompose** each problem by following explicit
    textual cues (headings, first second, to this end, method
    overviews, figure/table captions, bullet lists, etc.).
3. **Identify leaves**: a node is a leaf *only if* it describes a
    concrete experiment and you can cite the corresponding Figure /
    Table / Section ID.
4. **Capture all layers**do **not** skip intermediate hypotheses,
    objectives, or analysis steps the paper explicitly discusses.
5. **Stay faithful** to the papers wording for technical terms;
    paraphrase only for brevity or clarity.
6. **No outside invention**derive every node from the paper alone.
    If information is missing, mark the node with [uncertain].
```

**Paper Parsing Prompt (Cont.)**

```
Strictly output the tree in a JSON format:
```
{ "paper": {
    "title": "",
    "authors": [],
    "venue": "",
    "year": ""
  },
  "problem_tree": {
    "node": "Root: broadest research problem tackled by the paper",
    "type": "root node",
    "description": "a detailed description of the research problem
    in this node",
    "evidence": "references back to the original paper to back up
    the construction of this node",
    "children": [
      {"node": "Intermediate sub-problem / objective 1",
        "type": "depth-1 node",
        "description": "a detailed description of the research
    problem in this node",
        "evidence": "references back to the original paper to back
    up the construction of this node",
        "children": [
          {
            "node": "Narrower question or method component",
            "type": "depth-2 node",
            "description": "a detailed description of the research
    problem in this node",
            "evidence": "references back to the original paper to
    back up the construction of this node",
            "children": [
              {"type": "leaf node",
                "task": "Concrete experimental task (as phrased by
    paper)",
                "dataset": ["..."],
                "model_or_method": ["..."],
                "metrics": ["..."],
                "protocol_or_setup": "key settings/splits/
    hyperparams if stated",
                "evidence": {
                  "figure": "Fig. X",
                  "table": "Table Y",
                  "section": "Sec. Z or Result subsection name"
                },
                "conclusion": "explicit and detailed conclusions
    derived from experiments in this current leaf node",
                "status": ""  // leave empty or set to "[uncertain]"
     if any item is missing in the paper
              }
            ]
          }
        ]
      },
      {
        "node": "Intermediate sub-problem / objective 2",
        "children": [ /* ...more branches ending in leaves... */ ]
      }
    ]
  }
}
```

Table 5: Taxonomy of Agent Failure Modes in Experimental Workflows

| Phase | Error Type | Description |
|---|---|---|
| **Planning** | Unsound Design | Fails to formulate comprehensive experimental protocols, omitting critical methodological components or containing fundamental design flaws |
| | Disobey Task Specification | Deviates from provided instructions, executing unrelated or incorrect experimental procedures |
| **Implementation** | Implementation Error | Produces incomplete or faulty code containing critical bugs (e.g., missing data normalization, failed result extraction) |
| | Laziness | Employs insufficient sample sizes or minimal computational effort, constraining statistical power and pattern detection |
| **Execution** | Premature Termination | Halts after script generation or planning phases without executing the experimental pipeline |
| | Endless Loop | Fails to recognize termination conditions, perpetually re-attempting conclusions or initiating redundant experiments |
| **Analysis** | Interpretation Error | Correctly executes experiments but misinterprets resulting data, overlooking critical patterns or trends |
| | Conclusion Error | Formulates unsubstantiated or overgeneralized claims unsupported by experimental evidence |
| **System** | System Error | Encounters infrastructure-level failures or runtime exceptions during operation |

---

**Research Input Example**

```
You are a research agent. Conduct research and experiment about the
    question: ""How do self-correction methods impact large language
    model performance across math, commonsense reasoning, and multi-
    hop question answering benchmarks?""

You have access to the following resources:

Models:
- gpt-3.5-turbo and gpt-4o via the provided inference utilities
- Use a temperature of 1
- You can call these models using: from utils.llm_inference import
    LLMInference
- API key is provided with the LLMInference initialization function
- You can use the batch_generate() function to speed up the
    experiment
- Computational budget: 300 API calls per model

Datasets:
- GSM8K (grade school math word problems): /data/gsm8k/
- CommonSenseQA (commonsense multi-choice QA): /data/commonsenseqa/
- HotpotQA (open-domain multi-hop QA): /data/hotpotqa/

Experimental constraints:
- Evaluate using the Exact Match accuracy metric

Please design and execute **FULL** experiments to investigate this
    research question. Document your experimental plan, run your
    experiments, and provide conclusions at different levels of
    detail.
```

