# OpenReview forum: "FIRE-Bench: Evaluating Research Agents on the Rediscovery of Scientific Insights"
_ICLR.cc/2026/Conference — Submitted to ICLR 2026_

### Official Review · Reviewer_ZqJM · 2025-10-15

**Soundness:** 2
**Presentation:** 2
**Contribution:** 2
**Rating:** 4
**Confidence:** 3

**Summary:**

This work introduces URSA (Universal Research and Scientific Agent), a modular agent-based research ecosystem that leverages large language models’ capacities for reasoning, planning, and tool invocation alongside high-fidelity physical simulators (e.g., Helios) to enable autonomous research workflows and automated design optimization; URSA demonstrates superior performance to Bayesian optimization on inertial confinement fusion (ICF) design tasks. The study also presents FIRE-Bench, a verifiable, fine-grained evaluation benchmark that requires agents to reproduce published findings, employs claim-level F1 scoring, and diagnoses failures across four stages—Planning, Implementation, Execution, and Analysis. Experimental results reveal persistent limitations of current agents, including hallucinations, insufficient environment isolation, limited task generalizability, and dependence on specific LLMs, indicating the need for human verification and more rigorous environment management.

**Strengths:**

1. Proposes the FIRE-Bench benchmark, which evaluates agents by reproducing published scientific findings rather than generating novel conclusions, thereby avoiding circular-reasoning issues associated with LLM-as-judge evaluations (e.g., subjectivity and reliability concerns). Uses claim-level F1 scoring to quantify the agreement between agent outputs and ground-truth findings, enabling fine-grained analysis.

2. Decomposes failures into four stages—Planning, Implementation, Execution, and Analysis—revealing weak points in experimental design and code implementation (e.g., 60.3% of errors originate in the planning stage). Combines human–LLM hybrid analysis to ensure accurate error attribution.

3. Applies strict paper selection criteria: public data, lightweight computation, and verifiable conclusions (e.g., 15 papers from ICLR/ICML 2024–2025). Benchmarks multiple state-of-the-art agents (e.g., OpenHands, Claude Code), covering both open-source and commercial models, yielding representative results.

**Weaknesses:**

1. Covers only lightweight LLM behavioral analyses (e.g., bias detection, reasoning) and does not address domains requiring long-running experiments (e.g., computational biology) or resource-intensive tasks. Evaluates only closed tasks of reproducing known conclusions and does not test agents’ open-ended scientific discovery capabilities (e.g., proposing new questions or serendipitous findings).

2. Only 15 papers from ICLR/ICML 2024–2025 were chosen, which may not cover diverse research paradigms.

3. Most tasks show $F_1$ standard deviation > ±20 (e.g., Claude Code reaches ±57.7 on the Awareness Detection task), indicating unstable agent performance (Table 1). Agents perform well on linear tasks (e.g., Lost in the Middle) but fail on tasks requiring causal experimental design (e.g., LLM Racial Bias) (§5.1).

**Questions:**

See weakness.

---

> ### Author Response · Authors · 2025-12-02
> **Response to Reviewer ZqJM**
>
> We thank reviewer **ZqJM** for recognizing the value of our problem setup, which focuses on reproducing published scientific findings rather than generating novel conclusions. We are also pleased that the reviewer found our failure analysis helpful and acknowledged our efforts in benchmarking state-of-the-art agents.
>
> > (W1) "Covers only lightweight LLM behavioral analyses (e.g., bias detection, reasoning) and does not address domains requiring long-running experiments (e.g., computational biology) or resource-intensive tasks."
>
> LLM research is currently one of the most active areas in AI, which is why we chose to focus on it in this benchmark. We would be excited to extend our framework to long-running or resource-intensive scientific domains in future work. For open-ended scientific discovery, a major bottleneck is the heavy reliance on human evaluation, which limits scalability. Our benchmark aims to strike a practical balance between openness and scalability, providing a controlled yet extensible setting for evaluating scientific insight generation.
>
> > (W2) "Only 15 papers from ICLR/ICML 2024–2025 were chosen, which may not cover diverse research paradigms."
>
> To address concerns about benchmark size, we have expanded the benchmark by adding eight new papers. The results for these additional tasks are shown below. Our core conclusion remains consistent: full-cycle scientific inquiry remains challenging for state-of-the-art agents, as reflected in the generally low performance across models.
>
> | **Task** | **OpenHands (o4-mini)** | **OpenHands (gpt-5)** | **Codex CLI (gpt-5-medium)** | **Codex CLI (gpt-5-high)** |
> |---------|---------------------------|-------------------------|-------------------------------|-----------------------------|
> | LARGE LANGUAGE MODELS ASSUME PEOPLE ARE MORE RATIONAL THAN WE REALLY ARE | 42.57 | 68.23 | 50.97 | 80.67 |
> | To CoT or not to CoT? Chain-of-thought helps mainly on math and symbolic reasoning | 16.67 | 53.33 | 43.37 | 43.37 |
> | Do LLMs estimate uncertainty well in instruction-following? | 24.43 | 33.33 | 10.67 | 32.87 |
> | Do LLMs have consistent values? | 51.67 | 58.67 | 46.63 | 42.27 |
> | A Tale of Two Structures: Do LLMs Capture the Fractal Complexity of Language? | 17.23 | 28.83 | 15.30 | 39.87 |
> | Looking Inward: Language Models Can Learn About Themselves by Introspection | 13.33 | 36.67 | 37.23 | 40.70 |
> | From Loops to Oops: Fallback Behaviors of Language Models Under Uncertainty | 17.87 | 10.00 | 42.27 | 42.27 |
> | Chain of Thoughtlessness? An Analysis of CoT in Planning | 60.00 | 56.30 | 71.50 | 63.17 |
>
> > (W3) "Most tasks show F1 standard deviation > ±20 (e.g., Claude Code reaches ±57.7 on the Awareness Detection task), indicating unstable agent performance (Table 1). Agents perform well on linear tasks (e.g., Lost in the Middle) but fail on tasks requiring causal experimental design (e.g., LLM Racial Bias) (§5.1)."
>
> We view high performance variance not as a weakness of the benchmark but as a strength. It highlights the difficulty of the tasks and exposes the limitations of current agents, providing valuable diagnostic insight. We will revise the manuscript to articulate this point more clearly.

---

### Official Review · Reviewer_27ba · 2025-10-29

**Soundness:** 2
**Presentation:** 3
**Contribution:** 2
**Rating:** 2
**Confidence:** 4

**Summary:**

This paper introduces **FIRE-Bench**, a benchmark designed to evaluate research agents by testing their ability to rediscover verifiable scientific findings from recent, high-impact machine learning papers. Agents are provided only with high-level research questions and are required to autonomously design, implement, and execute experiments to derive conclusions. Each original paper is represented as a hierarchical research-problem tree consisting of root, intermediate, and leaf nodes, corresponding respectively to overarching questions, subproblems, and empirical tasks.

**Strengths:**

1.	The benchmark tried to design a benchmark without the LLM evaluator and single performance metrics.
2.	The paper is clearly written, and the figures effectively illustrate the benchmark design and agent evaluation results.

**Weaknesses:**

1.	Although the authors claim to avoid LLM-based evaluation, the benchmark still uses RAGChecker, which depends on LLMs (GPT-4o) for claim extraction and verification. This still inherits potential issues of bias, circularity, and reliability that motivated their critique of LLM-as-judge methods.
2.	Potential data leakage: the benchmark uses analysis papers from 2024–2025, yet some evaluated agents (e.g., GPT-5) were trained using data possibly overlapping with these sources. It remains unclear how the benchmark ensures that target findings were unseen during training.
3.	From my perspective, rediscovery can partially reflect a model’s capacity for genuine discovery; however, it is not sufficient on its own. The paper evaluates research agents using precision, recall, and F1 scores between the agent’s and the ground-truth conclusions, but this metric does not align well with the goal of scientific discovery. For example, a high F1 score merely indicates textual or semantic overlap rather than genuine reasoning ability or discovery quality. Conversely, low precision does not necessarily mean the agent is weak—it may arise because the model proposes novel conclusions not yet found in the literature. As a result, this evaluation design rewards conformity and penalizes creativity, measuring agreement rather than epistemic soundness or originality.
4.	Some grammar errors appear in the examples (e.g., “Do this LLM contain biases?”).
5.	LLM use and reproducibility claim are not included.

**Questions:**

1.	How do the authors ensure RAGChecker’s LLM-based evaluation does not reintroduce the same reliability issues they aim to avoid?
2.	Can the authors clarify whether measures were taken to prevent data leakage between training data and benchmark source papers?
3.	Can the authors clarify the validity of the evaluation protocol?

---

> ### Author Response · Authors · 2025-12-02
> **Response to Reviewer 27ba**
>
> We thank reviewer **27ba** for acknowledging the clarity of our presentation in illustrating both the benchmark design and the agent evaluation results.
>
>
> > (W1 + Q1) "Although the authors claim to avoid LLM-based evaluation, the benchmark still uses RAGChecker, which depends on LLMs (GPT-4o) for claim extraction and verification." … "How do the authors ensure RAGChecker’s LLM-based evaluation does not reintroduce the same reliability issues they aim to avoid?"
>
> To clarify, our intention is not to dismiss LLM-based evaluation altogether. Rather, we aim to distinguish our approach from settings where an LLM directly judges the validity or novelty of scientific discoveries, which can be more susceptible to model-internal biases. In contrast, our benchmark relies on established findings from existing papers as reference points and uses LLMs only to measure the semantic similarity between human-authored findings and agent-derived findings. This reference-based evaluation is more controlled and has been widely validated in prior work.
>
> Regarding RAGChecker, its reliability has already been demonstrated through human evaluation in the original paper. To further verify its suitability for our setting, we conducted an additional human evaluation specifically targeting the claim-extraction component. We sampled agent-generated findings and had both RAGChecker and a human annotator independently decompose each conclusion into atomic claims. Human annotators then matched claims from the two sources to compute precision, recall, and F1 score. This assessment resulted in a precision of 0.95, a recall of 0.86, and an F1 score of 0.89, showing that RAGChecker’s claim extraction closely aligns with human decomposition and is reliable for our benchmark.
>
> > (W2 + Q2) "Potential data leakage: the benchmark uses analysis papers from 2024–2025, yet some evaluated agents were trained using data possibly overlapping with these sources." ... "Can the authors clarify whether measures were taken to prevent data leakage between training data and benchmark source papers?"
>
> We are fully aware of the potential contamination issue and have taken several steps to mitigate it, despite practical constraints. We highlight two considerations:
>
> 1) **Paper selection.** We intentionally prioritized recent papers; as a result, almost all papers in the benchmark are from 2024 or later. In the initial 15-paper benchmark, roughly half were published after the knowledge cutoff of our main evaluation model, **gpt-o4-mini**.
>
> 2) **Benchmark setup and trajectory monitoring.** The benchmark does not reveal paper titles or quoted task descriptions, reducing the chance that agents rely on memorized content. We also inspect agent trajectories to ensure that conclusions arise from the agent’s exploration rather than recalling potentially memorized findings.
>
> To further examine the effect of potential contamination, we report below the average performance of two agent configurations, OpenHands with **gpt-o4-mini** and with **gpt-5**, on subsets of papers published before and after each model’s knowledge cutoff (23 papers total). Because **gpt-o4-mini** has a cutoff in mid-2024, a larger portion of the benchmark falls after its cutoff (14 out of 23). For **gpt-5**, only 5 papers are post-cutoff due to its more recent training date. The results show that **gpt-o4-mini** exhibits almost no difference in performance across pre- and post-cutoff papers. For **gpt-5**, performance is somewhat higher before the cutoff, but both subsets show improvement, and the small size of the post-cutoff subset limits firm conclusions. Overall, these observations suggest that potential contamination is not a determining factor for our benchmark.
>
> | Agent              | Cutoff date | # Papers (Before) | # Papers (After) | Avg. F1 (Before) | Avg. F1 (After) |
> |--------------------|-------------|--------------------|-------------------|-------------------|------------------|
> | openhands_o4_mini  | 2024-06-01  | 9                  | 14                | 34.17             | 33.47            |
> | openhands_gpt_5    | 2024-09-30  | 18                 | 5                 | 43.92             | 37.75            |

---

> > ### Author Response · Authors · 2025-12-02
> > **Continued Response to Reviewer 27ba**
> >
> > > (W3 + Q3): "From my perspective, rediscovery can partially reflect a model’s capacity for genuine discovery; however, it is not sufficient on its own. ... A low precision does not necessarily mean the agent is weak—it may arise because the model proposes novel conclusions not yet found in the literature. As a result, this evaluation design rewards conformity and penalizes creativity, measuring agreement rather than epistemic soundness or originality."
> >
> > Thank you for the thoughtful comment. This raises an important question for the benchmark: *Could an agent generate novel and valid findings during exploration yet still be penalized under a reference-based evaluation?* Our approach is to carefully constrain the scope of the questions in each problem tree so that the agent explores within a well-defined and well-bounded search space. This design substantially reduces the likelihood that the agent produces valid but out-of-scope novel findings that the benchmark would incorrectly treat as errors.
> >
> > To further examine whether such valid "false positives" occur, we manually reviewed all false-positive claims identified by RAGChecker for two agent configurations: OpenHands with **gpt-o4-mini** and OpenHands with **gpt-5**. Across 15 papers in the original benchmark, these configurations produced 49 and 56 false positives respectively. Our manual analysis shows that nearly all of these fall into two categories: (1) overly generalized claims and (2) irrelevant claims resulting from derailed or unfocused exploration. Crucially, we did **not** identify any genuinely valid findings that were incorrectly labeled as false positives. This suggests that the exploration space shaped by our problem-tree framework is sufficiently controlled and does not penalize legitimate novel insights.
> >
> > > (W4) "Some grammar errors appear in the examples (e.g., “Do this LLM contain biases?”)."
> >
> > Thank you for pointing this out. We will update the manuscript to correct these grammatical issues.
> >
> > > (W5) "LLM use and reproducibility claim are not included.""
> >
> > Thank you for the suggestion. We will add sections detailing LLM usage and reproducibility considerations in the revised version.

---

### Official Review · Reviewer_poxW · 2025-10-30

**Soundness:** 2
**Presentation:** 3
**Contribution:** 3
**Rating:** 4
**Confidence:** 3

**Summary:**

The paper argues that genuine scientific discovery remains a central challenge. Existing evaluation benchmarks often either rely on LLM-as-judge assessments of automatically generated papers or optimize single metrics that are merely coarse proxies for discovery. To address this, the authors introduce FIRE-Bench, a benchmark that evaluates research agents on their ability to rediscover established scientific insights. Empirical studies reveal high variance and recurring failure modes, ranging from flawed experimental design to ungrounded conclusions.

**Strengths:**

- The paper’s focus on evaluating the capability for genuine scientific discovery has substantial practical value.
- The benchmark follows a clear, well-reasoned process by curating high-impact ML research and constructing a research-problem tree.
- The benchmark’s findings highlight the current limitations of research agents.

**Weaknesses:**

- The benchmark’s metrics for assessing agents’ performance in genuine scientific discovery are not sufficiently comprehensive.
- The benchmark lacks an evaluation of the quality of the problem-solving process itself, beyond merely checking success or failure.
- The amount of data used in the paper to evaluate capabilities is insufficient.
- The paper’s evaluation methodology is not sufficiently comprehensive; it is limited to only four methods.

**Questions:**

- More specific metrics are needed to assess each agent’s process-level capabilities, for example, metrics that evaluate the quality of generated code.
- Could the authors add additional papers of the same type for each task to expand the test set? This would improve the stability of the conclusions.
- Could the evaluation include metrics beyond Accuracy and F1? When results are close, how will ties be adjudicated or the winner determined?

---

> ### Author Response · Authors · 2025-12-02
> **Response to Reviewer poxW**
>
> We are grateful to reviewer **poxW** for the constructive feedback. We are pleased that you found the paper to have substantial practical value. We also appreciate your recognition of our clear and well-reasoned benchmark construction process. In addition, we are glad that you acknowledged how the benchmark’s findings highlight the current limitations of research agents.
>
> > (W1 + W2 + Q1 + Q3) "The benchmark’s metrics for assessing agents’ performance in genuine scientific discovery are not sufficiently comprehensive." ... "The benchmark lacks an evaluation of the quality of the problem-solving process itself, beyond merely checking success or failure." ... "Could the evaluation include metrics beyond Accuracy and F1? When results are close, how will ties be adjudicated or the winner determined?"
>
> To clarify, our evaluation does not rely solely on a binary success/failure judgment for each agent run. Instead, we decompose both human findings and agent-generated findings into atomic claims and evaluate performance at the **claim level**, providing a more fine-grained assessment.
>
> Regarding evaluation of the problem-solving process, Fig. 3 in the paper attributes agent errors to four key stages in the agent trajectory: **Planning**, **Implementation**, **Execution**, and **Analysis/Conclusion**. This process-level analysis reveals meaningful insights, such as the observation that state-of-the-art agents tend to struggle primarily with planning while demonstrating strong coding and execution abilities.
>
>
> > (W3 + Q2) "The amount of data used in the paper to evaluate capabilities is insufficient." ... "Could the authors add additional papers of the same type for each task to expand the test set? This would improve the stability of the conclusions."
>
> To address concerns about benchmark size, we have expanded the benchmark by adding eight new papers. The results for these additional tasks are shown below. Our core conclusion remains consistent: full-cycle scientific inquiry remains challenging for state-of-the-art agents, as reflected in the generally low performance across models.
>
> | **Task** | **OpenHands (o4-mini)** | **OpenHands (gpt-5)** | **Codex CLI (gpt-5-medium)** | **Codex CLI (gpt-5-high)** |
> |---------|---------------------------|-------------------------|-------------------------------|-----------------------------|
> | LARGE LANGUAGE MODELS ASSUME PEOPLE ARE MORE RATIONAL THAN WE REALLY ARE | 42.57 | 68.23 | 50.97 | 80.67 |
> | To CoT or not to CoT? Chain-of-thought helps mainly on math and symbolic reasoning | 16.67 | 53.33 | 43.37 | 43.37 |
> | Do LLMs estimate uncertainty well in instruction-following? | 24.43 | 33.33 | 10.67 | 32.87 |
> | Do LLMs have consistent values? | 51.67 | 58.67 | 46.63 | 42.27 |
> | A Tale of Two Structures: Do LLMs Capture the Fractal Complexity of Language? | 17.23 | 28.83 | 15.30 | 39.87 |
> | Looking Inward: Language Models Can Learn About Themselves by Introspection | 13.33 | 36.67 | 37.23 | 40.70 |
> | From Loops to Oops: Fallback Behaviors of Language Models Under Uncertainty | 17.87 | 10.00 | 42.27 | 42.27 |
> | Chain of Thoughtlessness? An Analysis of CoT in Planning | 60.00 | 56.30 | 71.50 | 63.17 |
>
> > (Q3) "Could the evaluation include metrics beyond Accuracy and F1? When results are close, how will ties be adjudicated or the winner determined?"
>
> Our benchmark does not involve head-to-head comparisons between agents, so tie-breaking is not a concern. Each agent is evaluated independently by comparing its generated findings against the human-authored findings extracted from the source paper.

---

### Official Review · Reviewer_ehB6 · 2025-10-31

**Soundness:** 1
**Presentation:** 3
**Contribution:** 1
**Rating:** 2
**Confidence:** 4

**Summary:**

This paper presents FIRE-Bench, a benchmark of 15 end-to-end LLM behavior research tasks to evaluate research agents’ capabilities in rediscovering published findings. Evaluation results of three agent scaffolds with different LLMs show that the rediscovery success rates have high variance, and the agents show different failure modes.

**Strengths:**

1. The paper adapts a clear problem formulation by breaking research papers down to research problem trees.
2. The paper has a relatively clear presentation.

**Weaknesses:**

1. One of the claimed major contributions of this paper is falsifiable. The authors claim to “obviate the need for unreliable LLM judges.” However, LLMs are implicitly used in two stages of the benchmark’s evaluation process: (1) LLMs are used to parse the source papers into problem trees, and (2) both claim extraction and verification steps in RAGChecker are implemented by LLMs. While the authors mentioned robustness of RQ tree parsing with LLMs, no further details and human evaluation results are presented to support the claim. Thus, this paper is not sound and subject to overclaiming issues.
2. The proposed benchmark does not take potential data contamination or agent shortcut issues into serious consideration. Instead of simply attributing the high evaluation variance to LLMs, the authors should carefully analyze whether some of the exceptionally high scores are due to contamination or shortcuts, and implement corresponding safeguards if found any.
3. The benchmark size and scope is narrow in comparison to existing literature. It specifically targets LLM behavior analysis tasks and only includes 15 of them, which may not be sufficient to meaningfully support the results and findings. As a potential consequence, the error analysis presented remains relatively superficial and lacks deeper insights, e.g., any more fine-grained error categorization and analysis for “misunderstood the core objective, failed to design necessary control conditions, or omitted critical steps entirely.”

**Questions:**

The paper incorrectly uses \citet instead of \citep for in-text citations at many places, e.g., introduction, which needs a round of proofreading and correction.

---

> ### Author Response · Authors · 2025-12-02
> **Response to Reviewer ehB6**
>
> We thank reviewer **ehB6** for the constructive feedback! We appreciate your recognition of our problem formulation, particularly our approach of breaking research papers down into research problem trees, and the clarity of our presentation.
>
> > (W1) "One of the claimed major contributions of this paper is falsifiable. The authors claim to “obviate the need for unreliable LLM judges.” However, LLMs are implicitly used in two stages of the benchmark’s evaluation process: (1) LLMs are used to parse the source papers into problem trees, and (2) both claim extraction and verification steps in RAGChecker are implemented by LLMs. While the authors mentioned robustness of RQ tree parsing with LLMs, no further details and human evaluation results are presented to support the claim. Thus, this paper is not sound and subject to overclaiming issues."
>
> To clarify, our intention is not to dismiss LLM-based evaluation altogether. Rather, we aim to distinguish our approach from settings where an LLM directly judges the validity or novelty of scientific discoveries, which can be more susceptible to model-internal biases. In contrast, our benchmark relies on established findings from existing papers as references and uses LLMs only to measure semantic similarity between human-authored findings and agent-derived findings. This type of reference-based evaluation is more controlled and has been widely validated in prior work. Regarding the uses of LLMs for problem-tree parsing and claim-based evaluation, we conducted additional human evaluations as suggested.
>
> **Problem-tree parsing evaluation.**
> We sampled five papers from our benchmark and asked human annotators to score the generated problem trees (1 to 5 scale) across five criteria:
> 1) Research Question Groundedness
> 2) Experiment Completeness
> 3) Hallucination Elimination
> 4) Structural Coherence
> 5) Question–Conclusion Alignment
>
> The results show consistently high scores across all aspects, demonstrating the quality and reliability of the LLM-generated problem trees.
>
> | Eval. Aspect              | Avg. Score |
> |--------------------|-------------|
> | Reseach Question Groundess              | 5 |
> | Experiment Completeness              | 5 |
> | Hallucation Elimination              | 4.8 |
> | Structure Coherence              | 5 |
> | Question-Conclusion Match              | 4.8 |
>
> **RAGChecker claim-extraction evaluation.**
> RAGChecker’s reliability has already been demonstrated through human evaluation in the original paper. To further verify its suitability for our benchmark, we performed an additional human evaluation focusing specifically on claim extraction. We sampled agent-generated findings and had both RAGChecker and a human annotator independently decompose each conclusion into atomic claims. Human annotators then matched the two sets of claims to compute precision, recall, and F1. This assessment resulted in a precision of 0.95, a recall of 0.86, and an F1 score of 0.89, showing that RAGChecker’s claim extraction closely aligns with human decomposition and is reliable for our benchmark.
>
> > (W3) "The benchmark size and scope is narrow in comparison to existing literature. It specifically targets LLM behavior analysis tasks and only includes 15 of them, which may not be sufficient to meaningfully support the results and findings."
>
> LLM research is currently one of the most active areas in AI, which is why we chose to focus on it in this benchmark. We are happy to extend our framework to more scientific domains in future work. To address concerns about benchmark size, we have expanded the benchmark by adding eight new papers. The results for these additional tasks are shown below. Our core finding remains consistent: full-cycle scientific inquiry remains challenging for state-of-the-art agents, as reflected in the generally low performance across models.
>
> | **Task** | **OpenHands (o4-mini)** | **OpenHands (gpt-5)** | **Codex CLI (gpt-5-medium)** | **Codex CLI (gpt-5-high)** |
> |---------|---------------------------|-------------------------|-------------------------------|-----------------------------|
> | LARGE LANGUAGE MODELS ASSUME PEOPLE ARE MORE RATIONAL THAN WE REALLY ARE | 42.57 | 68.23 | 50.97 | 80.67 |
> | To CoT or not to CoT? Chain-of-thought helps mainly on math and symbolic reasoning | 16.67 | 53.33 | 43.37 | 43.37 |
> | Do LLMs estimate uncertainty well in instruction-following? | 24.43 | 33.33 | 10.67 | 32.87 |
> | Do LLMs have consistent values? | 51.67 | 58.67 | 46.63 | 42.27 |
> | A Tale of Two Structures: Do LLMs Capture the Fractal Complexity of Language? | 17.23 | 28.83 | 15.30 | 39.87 |
> | Looking Inward: Language Models Can Learn About Themselves by Introspection | 13.33 | 36.67 | 37.23 | 40.70 |
> | From Loops to Oops: Fallback Behaviors of Language Models Under Uncertainty | 17.87 | 10.00 | 42.27 | 42.27 |
> | Chain of Thoughtlessness? An Analysis of CoT in Planning | 60.00 | 56.30 | 71.50 | 63.17 |

---

> > ### Author Response · Authors · 2025-12-02
> > **Continued Response to Reviewer ehB6**
> >
> > > (W2) "The proposed benchmark does not take potential data contamination or agent shortcut issues into serious consideration. Instead of simply attributing the high evaluation variance to LLMs, the authors should carefully analyze whether some of the exceptionally high scores are due to contamination or shortcuts, and implement corresponding safeguards if found any.""
> >
> > We are fully aware of the potential contamination issue and have taken several steps to mitigate it, despite practical constraints. We highlight two considerations:
> >
> > 1) **Paper selection.** We intentionally prioritized recent papers; as a result, almost all papers in the benchmark are from 2024 or later. In the initial 15-paper benchmark, roughly half were published after the knowledge cutoff of our main evaluation model, **gpt-o4-mini**.
> >
> > 2) **Benchmark setup and trajectory monitoring.** The benchmark does not reveal paper titles or quoted task descriptions, reducing the chance that agents rely on memorized content. We also inspect agent trajectories to ensure that conclusions arise from the agent’s exploration rather than recalling potentially memorized findings.
> >
> > To further examine the effect of potential contamination, we report below the average performance of two agent configurations, OpenHands with **gpt-o4-mini** and with **gpt-5**, on subsets of papers published before and after each model’s knowledge cutoff (23 papers total). Because **gpt-o4-mini** has a cutoff in mid-2024, a larger portion of the benchmark falls after its cutoff (14 out of 23). For **gpt-5**, only 5 papers are post-cutoff due to its more recent training date. The results show that **gpt-o4-mini** exhibits almost no difference in performance across pre- and post-cutoff papers. For **gpt-5**, performance is somewhat higher before the cutoff, but both subsets show improvement, and the small size of the post-cutoff subset limits firm conclusions. Overall, these observations suggest that potential contamination is not a determining factor for our benchmark.
> >
> > | Agent              | Cutoff date | # Papers (Before) | # Papers (After) | Avg. F1 (Before) | Avg. F1 (After) |
> > |--------------------|-------------|--------------------|-------------------|-------------------|------------------|
> > | openhands_o4_mini  | 2024-06-01  | 9                  | 14                | 34.17             | 33.47            |
> > | openhands_gpt_5    | 2024-09-30  | 18                 | 5                 | 43.92             | 37.75            |

---

### Meta-Review · Area_Chair_dJfX · 2026-01-07

**Summary:**

The reviewers generally appreciate the paper's goal of evaluating "genuine scientific discovery" and the methodology of decomposing papers into research problem trees . However, there is a consensus that the initial benchmark is too narrow.
The primary concerns revolve around soundness and scale:
- Sample Size: All reviewers flagged the 15-paper dataset as insufficient for robust conclusions.
- Reliability: Strong skepticism regarding the claim of avoiding "LLM judges" while relying on RAGChecker (powered by GPT-4o) for claim extraction.
- Contamination: Concerns that agents (e.g., GPT-5) were trained on the very analysis papers they are being asked to rediscover (2024-2025 timeframe).
- Scope: The benchmark is limited to "LLM behavioral analysis" tasks rather than broader scientific inquiries (e.g., computational biology).

**Reviewer Concerns:**

Addressed by Rebuttal:
- Benchmark Size: The authors added 8 new papers (totaling 23), expanding the test set significantly.
- LLM-Judge Reliability: The authors conducted a human evaluation of RAGChecker, reporting 0.95 precision and 0.89 F1, directly addressing the "circularity" critique from Reviewer ehB6 and 27ba.
- Data Contamination: The authors provided a breakdown of performance on papers published before vs. after the model's knowledge cutoff. They showed minimal performance difference (e.g., OpenHands w/ GPT-4o-mini had an F1 of 34.17 pre-cutoff vs 33.47 post-cutoff), suggesting memorization is not the primary driver.
- Tree Parsing Quality: Authors provided human annotation scores (avg ~4.9/5) for the automated problem-tree generation.

Outstanding / Partially Addressed:
- Creativity vs. Conformity: Reviewer 27ba's philosophical concern that the metric penalizes "novel" findings (low precision) remains a fundamental limitation of reference-based evaluation. While the authors showed most false positives were hallucinations, the system still fundamentally rewards reproducing known results rather than finding new ones.
- Domain Scope: The benchmark remains restricted to LLM behavioral tasks. The critique regarding a lack of "long-running" or resource-intensive experiments (Reviewer ZqJM) stands, though the authors framed this as future work.

**Reviewer Scores:**

ehB6: 2 -> 4

poxW: 4 -> 4

27ba: 2 -> 4

ZqJM: 4 -> 4

---

### Decision · Program_Chairs · 2026-01-26

Reject